# Association of Clinical Aspects and Genetic Variants with the Severity of Cisplatin-Induced Ototoxicity in Head and Neck Squamous Cell Carcinoma: A Prospective Cohort Study

**DOI:** 10.3390/cancers15061759

**Published:** 2023-03-14

**Authors:** Ligia Traldi Macedo, Ericka Francislaine Dias Costa, Bruna Fernandes Carvalho, Gustavo Jacob Lourenço, Luciane Calonga, Arthur Menino Castilho, Carlos Takahiro Chone, Carmen Silvia Passos Lima

**Affiliations:** 1Laboratory of Cancer Genetics, Faculty of Medical Sciences, University of Campinas, Campinas 13083-970, SP, Brazil; ligia.macedo@alumni.harvard.edu (L.T.M.);; 2Department of Anesthesiology, Oncology and Radiology, Faculty of Medical Sciences, University of Campinas, Campinas 13083-888, SP, Brazil; 3Department of Ophthalmology and Otorhinolaryngology, Faculty of Medical Sciences, University of Campinas, Campinas 13083-888, SP, Brazil

**Keywords:** cisplatin, ototoxicity, single-nucleotide variants, detoxification, DNA repair, apoptosis

## Abstract

**Simple Summary:**

Cisplatin is recognized as the standard agent for head and neck squamous cell carcinoma therapy, despite the relevant risk of permanent hearing damage. The aim of this study was to evaluate the possible associations of the clinicopathological features and inherited genotypes encoding cisplatin metabolism in eighty-nine patients undergoing chemoradiation with the risk of hearing loss. We were able to confirm race, body mass index, and cumulative cisplatin dose as independent clinical risk factors. Patients with specific isolated and combined genotypes encoding cisplatin efflux (*GSTM1*, *GSTP1* c.313A>G), DNA repair (*XPC* c.2815A>C, *XPD* c.934G>A, *EXO1* c.1762G>A, *MSH3* c.3133A>G), and apoptosis-related proteins (*FASL* c.-844A>T, *P53* c.215G>C) presented up to 32.22 higher odds of moderate or severe ototoxicity. These findings reinforce the importance of inherited nucleotide variants involved in cisplatin metabolism as candidate variables for predictive models of adverse events.

**Abstract:**

Background: Cisplatin (CDDP) is a major ototoxic chemotherapy agent for head and neck squamous cell carcinoma (HNSCC) treatment. Clinicopathological features and genotypes encode different stages of CDDP metabolism, as their coexistence may influence the prevalence and severity of hearing loss. Methods: HNSCC patients under CDDP chemoradiation were prospectively provided with baseline and post-treatment audiometry. Clinicopathological features and genetic variants encoding glutathione S-transferases (GSTT1, GSTM1, GSTP1), nucleotide excision repair (XPC, XPD, XPF, ERCC1), mismatch repair (MLH1, MSH2, MSH3, EXO1), and apoptosis (P53, CASP8, CASP9, CASP3, FAS, FASL)-related proteins were analyzed regarding ototoxicity. Results: Eighty-nine patients were included, with a cumulative CDDP dose of 260 mg/m^2^. Moderate/severe ototoxicity occurred in 26 (29%) patients, particularly related to hearing loss at frequencies over 3000 Hertz. Race, body-mass index, and cumulative CDDP were independent risk factors. Patients with specific isolated and combined genotypes of *GSTM1*, *GSTP1* c.313A>G, *XPC* c.2815A>C, *XPD* c.934G>A, *EXO1* c.1762G>A, *MSH3* c.3133A>G, *FASL* c.-844A>T, and *P53* c.215G>C SNVs had up to 32.22 higher odds of presenting moderate/severe ototoxicity. Conclusions: Our data present, for the first time, the association of combined inherited nucleotide variants involved in CDDP efflux, DNA repair, and apoptosis with ototoxicity, which could be potential predictors in future clinical and genomic models.

## 1. Introduction

Head and neck squamous cell carcinoma (HNSCC) is the sixth most common cancer worldwide, with 878,348 new cases and 444,347 deaths estimated in 2020 [1,2]. Approximately 75% of patients with HNSCC present locally advanced disease at diagnosis, and the standard therapy for most cases involves chemoradiation or the induction of multi-agent chemotherapy, in which cisplatin (CDDP) is usually included [3,4]. Alternative treatments, such as carboplatin or cetuximab, were studied in the context of chemoradiation, though their equivalence regarding efficacy has yet to be validated by randomized trials [5]. Additionally, in patients with treatment-naïve metastatic disease or platinum-sensitive relapse, CDDP-based regimens are commonly used in clinical practice, with benefits in progression-free survival (PFS) and overall survival (OS) [6,7]. Nonetheless, CDDP is related to significant adverse events, such as nausea, vomiting, nephrotoxicity, hypersensitivity reactions, and ototoxicity [3,8,9]. Among these, hearing impairment is a current concern since there are, to date, no effective otoprotective measures, resulting in potentially permanent and quality-of-life-limiting damage [10,11].

Every year, one in five patients submitted to CDDP-based chemotherapy will suffer severe to profound hearing loss [10,12,13]. Regarding chemoradiation for HNSCC, major losses are described in higher frequencies, with reported pure-tone median threshold increases ranging from 9.52 to 25 decibels (dB) at 4 kilohertz (kHz) and 18.57 to 27.14 dB at 8 kHz [14,15]. This event has a major negative impact on the quality of life [16] and requires essential care regarding dosage management and the duration of therapy [17]. Despite the association of cumulative CDDP dose, history of noise exposure, and smoking as independent risk factors, the prevalence and intensity of hearing impairment are remarkably heterogeneous among patients with similar characteristics and regimens [18]. This finding indicates the involvement of unknown risk factors, with single-nucleotide variants (SNVs), on genes encoding proteins related to CDDP metabolism, being potential candidates for this risk [19,20].

Numerous proteins act in the mechanisms of CDDP cellular detoxification, as well as in the pathways of damage repair and apoptosis [21,22] (Figure 1). 

The detoxification of CDDP occurs mainly through its conjugation with glutathione, encoded by the Mu1 (*GSTM1*), Theta1 (*GSTT1*), and Pi1 (*GSTP1*) genes [23], in which the lack of functional proteins involved in this cascade may contribute to intracellular CDDP accumulation and cytotoxic effects [24]. The cytotoxic activity of CDDP is also attributed to its DNA binding, leading to the activation of repair mechanisms. The DNA lesion induced by CDDP can be removed through the nucleotide excision repair (NER) pathway [25], mediated by the xeroderma pigmentosum (*XPC, XPD*, and *XPF*) [26,27] and excision repair cross-complementation group 1 (*ERCC1*) genes [28], as well as by the mismatch repair (MMR) pathway, mediated through proteins encoded by MutL homolog 1 (*MLH1*) [29], MutS homolog 2 (*MSH2*) [30], MutS homolog 3 (*MSH3*), and exonuclease 1 (*EXO1*) genes [29]. If the repair is ineffective, apoptosis is mediated by proteins encoded by *P53*, Caspase 8 (*CASP8*), *CASP9*, *CASP3*, Fas cell surface death receptor (*FAS*), and Fas ligand (*FASL*) tumor necrosis factors [21,31]. Defects in these pathways may promote increased DNA damage and/or apoptosis, with greater potential for toxicity [32].

Genome-wide studies have described SNVs in acylphosphatase 2 (*ACYP2*), involved in calcium homeostasis [33,34,35] and Mendelian deafness *WFS1* genes [20,33,36,37], as predictors of CDDP-induced ototoxicity. Genes encoding thiopurine S- (*TPMT*) and cathecol-O methyltransferases (*COMT*) have also been described as potential risk factors [35]. In CDDP-treated patients, *GSTM1*, *GSTT1* [18,38,39,40,41,42], and *GSTP1* c.313A>G [38,39,41,43] were seen in pediatric solid or adult testicular tumors with controversial results in ototoxicity, while *XPC* c.2815A>C SNV influenced ototoxicity in osteosarcoma patients [44].

To our knowledge, the only cohort that evaluated SNVs in genes of distinct pathways of CDDP metabolism, damage repair, and apoptosis (*GSTM1*, *GSTT1*, *GSTP1* c.313A>G, *XPC* c.2815A>C, *XPD* c.934G>A, *XPD* c.2251A>C, *XPF* c.2505T>C, *ERCC1* c.354C>T, *MLH1* c.-93G>A, *MSH2* c.211 +9G>C, *MSH3* c.3133A>G, *EXO1* c.1762G>A, *P53* c.215G>C, *FAS* c.-671A>G, *FAS* c.-1378G>A, *FASL* c.-844C>T, *CASP3* c.-1191A>G, and *CASP3* c.-182-247G>T) in the ototoxicity of HNSCC treated with CDDP chemoradiation was previously conducted by our group, and the functional roles of each SNV described in the literature are presented in Table A1. We found that *GSTT1* [45], *EXO1* [19], *XPC* [46], and *FASL* [47] SNVs altered the occurrence of all-grade ototoxicity.

Since there is scarce information regarding pure tone and audiometric speech changes in patients under CDDP chemoradiation, considering that moderate/severe ototoxicity influences quality of life and the fact that patients may inherit defects in more than one pathway, we conducted a descriptive and pharmacogenetic study focusing isolated factors related to CDDP metabolism, aiming to contribute to the prompt recognition of patients at high risk of ototoxicity before treatment initiation and thus enabling treatment modifications.

## 2. Materials and Methods

### 2.1. Study Population

This cohort prospectively enrolled HNSCC patients who were eligible for treatment with definitive chemoradiation at the Clinical Oncology Service of the University of Campinas, Brazil, between June 2011 and February 2014. Eastern Cooperative Oncology Group (ECOG) performance status of equal to or less than 1 [48], creatinine clearance greater than 45 mL/min, and the absence of baseline moderate or severe hearing impairment were required. Patients who were not candidates for treatment with CDDP or who were under induction, adjuvant, or palliative therapy were excluded.

Patients received high-dose CDDP (starting dose of 80–100 mg/m2 on days 1, 22, and 43) [49] associated with RT (35 sessions; planned total radiation dose of 70 Gray—Gy). All patients received anti-emetic prophylaxis with intravenous ondansetron and dexamethasone pre-infusion, in addition to oral dexamethasone and metoclopramide, for the following three days. Mannitol and hydration with saline solution, potassium chloride, and magnesium sulfate were administered, as reported [46]. Dose delays and reductions were applied in toxicity events with grades equal to or greater than 3, according to the National Cancer Institute criteria for adverse events (NCI CTCAE) [50]. Patients were followed from recruitment to 30 days after treatment completion.

The study was approved by the local institutional review board (Protocol 274/2011 and 62870722.1.0000.5404), and all patients enrolled in the study agreed to participate and declared consent in accordance with the Declaration of Helsinki. The results of this study were reported following the Strengthening the Reporting of Observational Studies in Epidemiology (STROBE) guidelines [51].

### 2.2. Clinical Data

Data related to age, gender, race, history of tobacco and alcohol use [52], ECOG status [48], body mass index (BMI) [53], and presence of diabetes [54] or systemic hypertension [55] as comorbidities of interest were collected. Regarding disease characteristics, primary tumor location, tumor side, and histological grade were also computed. Diagnosis and tumor staging followed the American Joint Committee on Cancer criteria [56]. Data related to cumulative CDDP dose, radiotherapy (RT) technique (2D or 3D), and final total dose in Gy, including total doses from supraclavicular fossa, cervico-facial, cervico-posterior, and boost, were also registered for analysis.

### 2.3. Hearing Assessment

Patients were submitted to otoscopic examination before any audiometric measurements. If there were identifiable diseases of the external acoustic meatus, tympanic membranes, middle ears, or other conditions that could interfere with the audiological evaluation, patients received treatment and were followed up until resolution. Audiometric evaluations were performed on two occasions, before treatment initiation and up to 30 days after therapy completion in an acoustic booth previously calibrated to meet the specifications of internal noise levels allowed according to the International Organization for Standardization ((ISO) 8253-1:2010 criteria, using the Interacoustics audiometer model AC 30 (Interacoustics A/S, Middlefart, Denmark).

#### 2.3.1. Pure Tone Audiometry

Pure tone audiometry was conducted in air and bone conduction for both the left and right sides. For the air conduction assessment, the tonal auditory thresholds were measured at sound frequencies 0.25, 0.50, 1, 2, 3, 4, 6, and 8 kHz, with earphones model TDH 39, applying the descending–ascending technique. In each test, the smallest sound stimulus perceived by the patient in at least 50% of the presentations was considered. Bone conduction evaluation was performed in a similar descending–ascending manner, registering minimum dB thresholds at frequencies 0.25, 0.50, 1, 2, 3, and 4 kHz through a conduction receiver bow fixed on the mastoid (Interacoustics A/S, Middlefart, Denmark). The corresponding hearing thresholds for each frequency in both ears were collected, considering that the normal expected range was lower than 15 dB [57].

#### 2.3.2. Speech Audiometry

Speech audiometry was performed when applicable, assessing the speech recognition threshold (SRT) in dB for the repetition of 50% disyllabic words for left and right ears. The speech discrimination score (SDS), calculating the percentage of syllables repeated correctly, was also registered for each side when possible [58]. SRT is normally within 10 dB of pure tone average thresholds, while the normal range of SDS is 92% to 100% [59]. Patients unable to speak owing to disease-related limitations or other causes had their exam halted and reasons noted.

### 2.4. Hearing Loss Classification

#### 2.4.1. Global Burden of Disease (GBD) Hearing Loss Classification

The GBD Hearing Loss Classification was performed, assessing the ISO threshold average for 0.5, 1, 2, and 4 kHz in dB, as recommended by the World Health Organization [60], pre- and post-treatment in air and bone conduction assessments [61]. Patients were categorized according to the criteria of unilateral (<20 dB in the better and >35 dB in the worst ears, respectively), mild (20 to 34 dB in the better ear), moderate (35–49 dB in the better ear), moderately severe (50–64 dB in the better ear), severe (65–79 dB in the better ear), and profound (80–94 dB in the better ear) hearing loss.

#### 2.4.2. National Cancer Institute Common Terminology Criteria for Adverse Events (NCI CTCAE)

Hearing loss in the right and left ears were also classified based on grades (G) 1 to 4, according to the NCI CTCAE v4.0 criteria [50] following the monitoring of at least 1, 2, 3, 4, 6, and 8 kHz audiogram, as follows: G 1, “threshold shift of 15 to 25 dB averaged at two contiguous test frequencies in at least one ear”; G 2, “threshold shift of >25 dB averaged at two contiguous test frequencies in at least one ear”; G 3, “threshold shift of >25 dB averaged at three contiguous test frequencies in at least one ear”; G 4, “decrease in hearing to profound bilateral loss (absolute threshold >80 dB hearing loss at 2 kHz and above)”.

### 2.5. Genetic Variants Analysis

Genetic variants were selected for study based on the National Center for Biotechnology Information database, minor allele frequency greater than 10%, previous association with risk/outcome of solid tumors and/or CDDP metabolism, and the availability of financial resources (Figure A1). For genotyping, DNA samples from peripheral blood were collected, where the genotypes of the *GSTM1* and *GSTT1* variants [62] were obtained through the multiplex polymerase chain reaction (PCR) followed by digestion assays with enzymes of restriction. The additional genetic variants were evaluated by real-time PCR using TaqMan^®^ SNP Genotyping Assays (Applied Biosystems^®^, Thermo Fisher Scientific Inc., Waltham, MA, USA), as follows: *GSTP1* c.313A>G (rs1695) [63], *XPC* c.2815A>C (rs2228001) [64], *XPD* c.934G>A (rs1799793) [65], *XPD* c.2251A>C (rs13181) [65], *XPF* c.2505T>C (rs1799801) [66], *ERCC1* c.354C>T (rs11615) [67], *MLH1* c.-93G>A (rs1800734) [68], *MSH2* c.211 +9G>C (rs2303426) [69], *MSH3* c.3133A>G (rs26279) [70], *EXO1* c.1762G>A (rs1047840) [71], P53 c.215G>C (rs1042522) [72], *FAS* c.-1378G>A (rs2234767) [73], *FAS* c.-671A>G (rs1800682) [74], *FASL* c.-844C>T (rs763110) [74], *CASP3* c.-1191A>G (rs12108497) [75], and *CASP3* c.-182-247G>T (rs4647601) [76]. Positive and negative controls were used in all genotyping reactions, and replications of 10% randomly selected samples were also performed in independent experiments, with 100% agreement.

### 2.6. Statistical Analysis

Descriptive statistics were performed according to the variables under study, with mean values and standard deviation (SD) in normal distribution or median and interquartile ranges (IQR) when applicable.

Pre- and post-treatment pure tone thresholds were described individually, as well as the averages of high-frequency minimum thresholds (considering 3, 4, 6, and 8 kHz) and ISO averages (0.5, 1, 2, and 4 kHz) in each ear. Wilcoxon’s signed-rank test for paired data was applied in the comparison of speech audiometry, pure tone averages, and frequency thresholds before and after chemoradiation, with the latter controlling for false discovery rates with the Benjamini–Hochberg correction in multiple testing [77]. Cochran’s Q test was used for the GBD classification of hearing loss before and after therapy. To assess the influence of clinicopathological aspects and genotypes related to high-frequency minimum threshold average changes from baseline, we performed multiple linear regression. Data were transformed into ranks. The significance level adopted for the study was 5%.

The main endpoint of this study was the proportion of patients with NCI CTCAE hearing loss G equal to or greater than 3 during follow-up based on audiometry monitoring. Multiple logistic regression was used to obtain the odds ratio (ORs) adjusted for any specific discrepancies for each independent variable, considering a 95% confidence interval (CI). Variables were selected using a conditional stepwise approach, permitting a *p*-value of under 0.10 in univariate regression. 

Post hoc power analyses (PA) were also conducted for associations, taking into consideration *p*-value and CI as the measures of statistical significance [78,79]. After multivariate analysis, results with *p* ≤ 0.05 were validated using bootstrap [80] to verify the stability of risk estimates and account for missing data (1000 replications). Isolated SNVs associated with the increase in the hearing thresholds or grade 3 ototoxicity and combined SNVs associated with an increase in hearing thresholds or grade 3 ototoxicity with PA >70% were selected for this study.

All tests were performed using the Statistical Analysis System (SAS) for Windows, version 9.4 (SAS Institute Inc., 2002–2008, Cary, NC, USA) and Stata Statistical Software: Release 15 (Stata Corp LP, College Station, TX, USA).

## 3. Results

### 3.1. Study Population

In a median follow-up of 142 days, 152 patients were enrolled, of whom 89 were included in the analysis with the completion of baseline and post-treatment audiometry (Figure 2).

The median age was 56 years, and most patients were male and white, with a high rate of tobacco and alcohol consumption. Median BMI was within the normally acceptable range, most presented an ECOG status of 0, and the proportion of diabetes and hypertension was 10 and 26.9%, respectively. Most primary tumors were in the oral cavity or oropharynx, evenly distributed between the right and left sides of the head and neck, well or moderately differentiated, and at advanced stages (III or IV). The median cumulative CDDP dose among patients was 260 mg/m^2^. Eighty-eight patients received 2D RT, with a total dose of 70 Gy. The clinicopathological aspects of patients enrolled in the study are further detailed in Table 1.

### 3.2. Hearing Impairment in Monitoring Audiometry

#### 3.2.1. Pure Tone Audiometry

Analyzing the median thresholds for each frequency upon baseline, we were able to observe a normal range below 2 kHz and a trend toward higher thresholds, starting from 3 kHz in both conduction modalities (Figure 3).

After treatment, there was significant damage regarding higher frequencies over 2 kHz, which was more evident in air conduction analysis. Hearing thresholds for each frequency in pure tone audiometry are further detailed in Table 2.

Following the ISO average criteria, a median increase of 5 dB (*p* < 0.001) on the right side and 6.25 dB (*p* < 0.001) on the left side was observed when comparing baseline to post-treatment assessments. For bone conduction, there was a median increase of 6.25 (*p* < 0.001) on the right side and 6.25 dB (*p* < 0.001) on the left side, respectively (Table 3).

Regarding high-frequency average thresholds for pure tone air conduction audiometry, there was a median increase of 18 dB (*p* < 0.001) on the right and 19 dB (*p* < 0.001) on the left sides observed after exposure to CDDP and RT.

#### 3.2.2. Speech Audiometry

Data from speech audiometry were retrievable in 62 patients since 27 had limited speech capability (nine were submitted to tracheostomy and eighteen presented tumors in the oral cavity). The median baseline SRT was 10 and 15 dB in the right and left ears, respectively. Pre- and post-treatment median differences were null on both sides (Table 3). Regarding SDS, median baseline and post-treatment scores were 96% and 92%, respectively, for both ears, with a decrease of 4% on the right side. 

#### 3.2.3. GBD Classification for Hearing Loss

Before chemoradiation, mild hearing loss was seen in about one-third and one-quarter of patients analyzed by air and bone conduction, respectively, and only three patients presented moderate hearing impairment in both assessments. After treatment, there was a significant increase in the proportion of mild and moderate hearing loss identified in air and bone conduction (chi^2^ 20.16, *p* < 0.001 for air; chi^2^ 18.24, *p* < 0.001 for bone conduction), although a severe degree of hearing loss was not observed throughout the study (Table 3). The unilateral loss was more evident in air conduction after treatment, although the same proportion was not observed in bone conduction analyses. All patients with unilateral damage after treatment had pharyngeal carcinoma located on the side of hearing loss and with changes at baseline.

#### 3.2.4. Hearing Impairment in Monitoring Audiometry According to NCI CTCAE Criteria

The proportion of any-grade hearing impairment by air conduction after chemoradiation was 76.4% (68 out of 89 patients). The ototoxicity of G1 and G2 was observed in 23 (25.8%) and 19 (21.3%) patients, respectively. G3 or moderate/severe ototoxicity occurred in 26 (29.3%) participants, and G4 was not identified in this study.

### 3.3. Clinicopathological Aspects and Genotypes in Hearing Impairment

#### 3.3.1. Average of Minimum Threshold for Pure Tone Air Conduction Audiometry at High Frequencies (3, 4, 6, and 8 kHz)

In univariate analysis, gender and cumulative CDDP dose were associated with hearing loss in the right ear, while BMI was associated with hearing loss in both ears. Only cumulative CDDP dose was associated with hearing loss in the right ear in multivariate analysis (regression coefficient = 0.08, *p* = 0.02), where the higher the dose of CDDP, the greater the hearing impairment (Appendix A).

When SNVs were analyzed individually, it was observed that patients with *XPC* c.2815AA genotype presented higher average threshold increases after CDDP chemoradiation than those with *XPC* c.2815AC or CC genotypes (23.8 versus 17.5 dB in the right ear; 27.5 versus 16.3 dB in left ear), as represented in Table 4. Higher average threshold increases were also seen after treatment in patients with combined genotypes *GSTM1* null plus *EXO1* c.1762GA or AA (21.3 versus 5.0 dB in the right ear; 22.5 versus 8.8 dB in the left ear) and with *GSTP1* c.313AG or GG plus *XPC* c.2815AA (30.0 versus 17.5 dB in the right ear; 38.8 versus 16.3 dB in the left ear) in comparison to other related variants. The analyses of isolated and combined SNVs with biological significance with hearing loss at high frequencies are presented in Appendix A, respectively.

#### 3.3.2. Hearing Impairment in Monitoring Audiometry According to NCI CTCAE Criteria

Race and BMI were significantly associated with the risk of G3 ototoxicity in univariate and multivariate analyses, but potential clinical risk factors such as age, gender, diabetes, hypertension, smoking, alcohol consumption, tumor stage, tumor side, and CDDP cumulative dose did not alter the risk of ototoxicity in univariate analysis (Appendix A). The occurrence of moderate/severe ototoxicity was more common in non-white than in white patients (66.7% versus 25.0%, respectively), with OR = 5.43 (95% CI: 1.21–24.27, *p* = 0.02) in multivariate analysis. BMI was also a potential predictor, as participants with grade 3 hearing impairment presented lower median BMI (17.8 versus 19.7), with the OR = 0.82 higher for every decrease in BMI (95% CI: 0.72–0.98, *p* = 0.03) in multivariate analysis.

When analyzed individually (Table 5), two SNVs were identified as independent factors for this outcome. Patients with *GSTP1* c.313AG or GG genotypes had about 4.20 higher odds of having grade 3 or greater ototoxicity. Moreover, *XPC* c.2815AA genotype was associated with greater odds of severe hearing impairment, with a reported OR of 3.13 (*p* = 0.01) in the multivariate regression model. In associations of SNVs, it was observed that patients with *GSTM1 null* plus the *XPC* c.2815AA genotype had 8.19 greater odds of having moderate/severe hearing impairment (*p* = 0.02, PA = 99%). *GSTP1* c.313AG or GG genotypes plus *XPC* c.2815AA, *XPD* c.934AA and *EXO1* c.1762AA had ORs of 32.22 (*p* = 0.004, PA = 97%), 19.44 (*p* = 0.02, PA = 92%), and 12.08 (*p* = 0.01, PA = 81%), respectively. In addition, we observed relevant associations amongst DNA repair and apoptosis-related SNVs in patients with *XPC* c.2815AA genotype plus *MSH3* c.3133A>G and *FASL* c.-844CC, where individuals with the respective profile had OR 17.09 (*p* = 0.009, PA = 88%) and 22.29 (*p* = 0.01, PA = 82%). Finally, patients with the combined genotypes *EXO1* c.1792GA or AA and *P53* c.215 CC had OR 20.97 (*p* = 0.02, PA = 85%). Further details of other SNVs and their combinations are summarized in Appendix A, respectively.

## 4. Discussion

In this clinical and pharmacogenetic cohort, it was possible to reaffirm the clinical relevance of hearing loss induced by CDDP. CDDP induces ototoxicity through the promotion of oxidative stress and inflammation in the cochlea, with the increased generation of reactive oxygen species (ROS) [81]. The long-term accumulation of CDDP in the cochlear endolymph was also described through plasma mass spectrometry in preclinical models, justifying the potential for permanent damage [82].

Firstly, substantial hearing loss before treatment was observed in our cohort; high-frequency minimum thresholds were higher at baseline, ranging from 35 to 40 dB over 4 kHz. This may be attributable to the high proportion of smokers in our sample since smoking is a reported risk factor for loss at high frequencies [18,83,84]. A history of noise exposure, not assessed in this cohort, could also explain this finding as well as uneven losses in left and right ears not associated with the tumor side [85,86,87]. We were able to observe a meaningful change after CDDP exposure in regard to minimum hearing thresholds, particularly in higher frequencies in univariate analysis, as suggested by previous studies [15], with limitations involving higher pitch sounds. Caballero and colleagues described similar findings in a cohort of 103 patients, with significantly meaningful changes after CDDP exposure (median change of 9.5 dB in the right and 18.75 dB in the left ears for 4 kHz; 18.6 dB in the right and 28.7 dB in the left, for 8 kHz). The limitations to quality of life related to hearing loss from CDDP have already been reported in a recent systematic review from Pearson and colleagues [16]. Regarding additional clinical factors, cumulative CDDP dose was observed as a risk factor for greater change in high-frequency averages (3, 4, 6, and 8 kHz) for the right ear, which prompted the inclusion of this variable in multivariate analysis for both sides.

When accounting for the 0.25 to 4 kHz interval, there was also a significant relative increase in mild and moderate hearing loss after CDDP in our analysis, following GBD classification. The percentage of 64.1% with a threshold ≥ 20 dB after treatment is markedly superior to the overall prevalence reported in the literature for the general population (19.3%) [88], pointing to the cytotoxic effects of CDDP on hearing impairment. To our knowledge, this is the first study to report this classification before and after CDDP in patients diagnosed with HNSCC [89]. Unilateral hearing damage was observed in four patients (4.5%) after therapy under pure tone audiometry air conduction, from which two (2.2%) had reported losses in both conduction modalities (air and bone). It is worth commenting that all patients with unilateral damage had pharyngeal carcinoma located on the side of hearing loss, and most had changes at baseline. Even though the RT technique and CDDP dose did not differ amongst patients, the location of the tumor in relation to the inner ear could have influenced this finding.

On the other hand, median outcomes from speech audiometry (SRT and SDS) were practically unchanged after platinum exposure. One possible explanation for this may be related to the fact that human speech usually ranges from 0.25 to 4 kHz [90], while CDDP-related hearing loss involves more relevant changes beyond 3 kHz. In an isolated acoustic environment, frequencies related to speech may be unaltered, though it is possible to expect a greater extent of limitation in terms of communication with background noise, which was not assessed in this cohort. The largest study analyzing speech audiometry after CDDP exposure was performed by Shahbazi and colleagues [91], evaluating the prevalence of speech recognition disability, defined as SRT greater than 15 dB, in testicular cancer survivors. In 1347 patients, speech recognition disability was identified in 10.4%, and the association of the cumulative CDDP dose could also not be confirmed. Those findings are distinct from our analysis, where 51.6% could be classified as speech-disabled before therapy and 60.7% after therapy. The study populations are markedly different since the Platinum Study [91] included younger patients not bearing primary tumors in the head and neck and without risk factors such as tobacco and alcohol consumption. There are, to date, only scarce amounts of the literature data on speech audiometry for HNSCC, thus limiting further comparisons. 

When considering the *NCI CTCAE* criteria for the classification of hearing loss, the proportion of 29.5% moderate/severe ototoxicity was marginally higher than previously reported literature data, ranging from 20 to 25% in adults [10,37]. Except for race and BMI, other clinical variables such as age, sex, tumor location, and staging could not be identified as prognostic factors in this analysis, and although cumulative CDDP is recognized as a risk factor for hearing damage [37], this association could not be observed in the present data for this outcome specifically. Some recent studies have suggested the presence of emotional stress as a possible risk factor for enhanced tumorigenesis and neurotoxicity induced by chemotherapy in general [92]. A cross-sectional analysis of 623 cancer survivors described a higher association of tinnitus (*p* = 0.029) and hearing loss (*p* = 0.007) amongst patients with higher distress scores [93]. Due to the characteristics of the study design, it is not possible to differentiate stress as an independent risk factor, as opposed to a consequence of long-term toxicity. Even though a longitudinal study of the current analysis could potentially assess this variable, distress scores were not previously planned and included in this cohort. 

In this study, *GSTP1* c313AG or GG and *XPC* c.2815 AA genotypes increased the odds of moderate/severe ototoxicity 4.20- and 3.13-fold, respectively. Preclinical studies have demonstrated that *GSTP1* c313 A>G encodes a change from isoleucine to valine in codon 105, leading to reduced protein activity and detoxification [63], while the XPC c.2815 C allele promotes the change from lysine to glutamine in codon 939, also diminishing protein activity and, consequently, DNA repair (Table A1) [64]. There is, however, marked heterogeneity of clinical effects in terms of the currently available literature. For instance, *GSTP1* c313AG or GG was associated with an increased risk of moderate/severe hearing impairment in 106 [41] and 64 children [43], respectively, treated with platinum agents, using the Brock hearing loss classification of equal or greater than 2 [20]. The association between cumulative CDDP and ototoxicity was found in the study conducted by Lui and colleagues [41] but not in the study by Sherief and colleagues [43]. Even though our findings in a previous analysis of the data [45] are similar and in agreement with the functional roles of *GSTP1* c313A>G [94], this SNV was not related to CDDP-induced ototoxicity in an additional cohort that recruited 71 children and young patients with various solid tumors [38], while in 173 patients with testicular carcinoma, post-treatment audiometric evaluations prompted divergent results, even though baseline assessments were not retrievable [39]. Reported results were also conflicting for isolated *XPC* c.2815A>G [35]. The *XPC* c.2815AA genotype was associated with an increased risk of any grade of toxicity [46] in a previous analysis of the data conducted by our group, and the same effect was observed in a smaller subset of patients with osteosarcoma [44]. Nonetheless, Lui and colleagues [41] did not present a significant association among 106 pediatric patients treated with platin analogs. Functional analyses performed for this variant [64] suggest the presence of the C allele reduces DNA repair, which would theoretically increase the risk of toxicity in contrast to what is currently reported, though an additional assay from Khan and colleagues [26] did not demonstrate a clear difference for the rate of nucleotide excision repair. Differences in the results obtained from the studies are not easily explained and may have originated from limitations related to sample size, patient baseline characteristics, tumor types, and treatment administered. There are also markedly distinct hearing loss classifications applied in previous cohorts, hampering proper direct comparisons with NCI CTCAE v4.03. Larger cohorts, in addition to further functional assays, would be ideal to better confront these findings.

The metabolism of CDDP is known to involve cellular efflux, NER, and MMR damage repair, as well as apoptosis [37]. We were able to observe meaningful interactions between variants encoding those distinct pathways, suggesting that toxicity may be enhanced by the coexistence of more than one mutation in the different stages of CDDP metabolism and cytotoxic effect. The combination of *GSTM1* null plus *XPC* c.2815AA, *GSTP1* c.313AG or GG plus *XPC* c.2815AA, *XPD* c.934AA or *EXO1* c.1762AA, or *XPC* c.2815AA plus *MSH3* c.3133AA or *FASL* c.-844CC (Table A1) intensified the odds of moderate/severe ototoxicity up to 32.22-fold. The variant alleles from SNVs *XPD* c.934 (A) [65], *EXO1* c.1762 (A) [71], and *MSH3* c.3133 (A) [70] have been shown to reduce DNA repair activity by encoding amino acid replacements with the consequent loss of protein function or expression (Table A1). Additionally, the SNV *FASL* c.-844 is located within the enhancer-biding region of *FASL*, and luciferase assays have described the variant genotype TT to promote protein affinity twice lower than wild CC donors, leading to less protein expression and, as such, reduced apoptosis [74]. Hence, the combination of genotypes enhancing CDDP accumulation and reducing repair or activating apoptosis could potentiate the risk of ototoxicity, as observed in this analysis. To our knowledge, no studies focusing on the effects of the combinations of SNVs on the genes of distinct pathways of CDDP metabolism have been conducted to date.

An association with MMR and apoptosis mechanisms was also noted in this study, as the combination of *EXO1* c.1762GA or AA and *P53* c.215CC genotypes increased the risk for events with OR 20.97. The *P53* c.215 wild allele encoding arginine (G) was described as more efficient in inducing apoptosis than the proline (C) variant [72]. The P53 protein signaling pathway promotes cell death triggered by the generation of ROS [35]. In addition to apoptosis, P53 is related to cell cycle arrest, cell senescence, and DNA repair [95]. Cellular senescence is a state of permanent cell cycle arrest that is able to promote local inflammation and tissue damage [96]. In vitro studies have suggested that early senescence in response to genotoxic stress was P53-dependent and EXO1-depleted [97,98]. Moreover, Benkafadar and colleagues [99] observed that the response to ROS-induced DNA damage leads to cochlear cell senescence by activating the P53 pathway and hence contributing to age-related hearing loss. To date, there is a lack of evidence for a direct association between ototoxicity by CDDP and cell senescence. However, the accumulation of senescent neuronal cells is associated with CDDP-induced peripheral neuropathy in mice [100]. Thus, we may infer that patients with *EXO1* c.1792GA or AA and *P53* c.215CC combined genotypes were more efficient in promoting cell cycle arrest and senescence of sensory cells after injury by CDDP and, consequently, were at greater risk of hearing loss when compared to patients carrying other genotypes.

We are aware that this study is limited for its sample size; thus, similarly to previous studies, lacking power for further SNVs combinations or polygenic evaluations and correction for other possible confounders. Though statistical tools were used to stabilize risk, such as bootstrap and power post hoc calculations, there may still be unknown influential factors not identifiable in this sample. It must also be considered that not all SNVs in the genes related to CDDP detoxification, DNA repair, and the apoptosis of damaged cells were evaluated in this study; only those recognized with a greater potential to induce ototoxicity were evaluated here. Thus, it is possible that other SNVs with equal or even greater importance in CDDP ototoxicity will be identified in future studies. Furthermore, other known SNVs for CDDP-induced ototoxicity unrelated to stages regarding drug absorption, distribution, metabolism, and excretion were not assessed and could be additional confounders. There is evidence supporting additional SNVs as risk factors for ototoxicity induced by CDDP related to *ACYP2* [33,34], *TPMT* [35], *COMT* [35], and *WFS1* [36] genes. *ACYP2* is known to influence ATP-dependent calcium signaling, which may play a role in sensorineural hearing loss [33]. TPMT and COMT are methyltransferases that may inactivate CDDP and purine compounds. The Mendelian deafness genes, amongst which *WSF1* is included, encode proteins reported to control endothelial reticulum stress response, thus influencing inner ear cellular damage [36]. Apart from known and unknown genetic risk factors for toxicity and hearing loss, clinical variables, such as a history of noise exposure [18] and distress scores [93], were also not collected from this cohort. We believe, however, that the exclusion of patients with reported hearing loss and hearing impairment in audiometry before treatment could, to some extent, attenuate these limitations.

It is also important to consider distinct patient characteristics when assessing the generalizability of this study for other tumors since the population was predominantly male, with a high frequency of smokers and alcohol users, as well as locally advanced stages of HNSCC. Treatment approaches in the field of RT may also be distinct and could influence the prevalence and severity of adverse events, mainly in institutions with more frequent use of intensity-modulated RT. Though prespecified treatment protocols were strictly followed, therefore preventing confounding to some extent, heterogeneity in therapy protocols could affect the generalizability of these results.

## 5. Conclusions

The results of this cohort suggest, for the first time, the interactions of inherited genetic abnormalities involved in CDDP metabolism as potential candidate targets for future risk models in ototoxicity. The development of genetic and clinical risk prediction tools is essential, not only for optimizing treatment selection based on efficacy but also to assist in supportive care during therapy. We believe these results may be included in future polygenic and clinical predictive models.

## Figures and Tables

**Figure 1 cancers-15-01759-f001:**
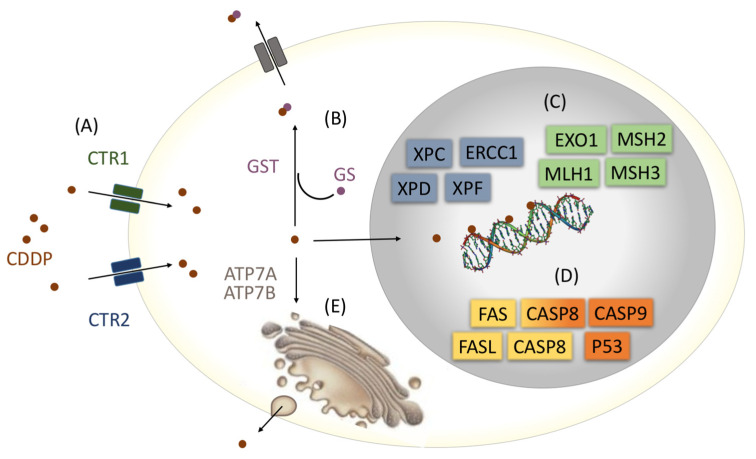
Main pathways related to cisplatin (CDDP) influx (A), detoxification (B), DNA repair (C), apoptosis (D), and efflux (E). CDDP influx occurs through copper transport receptors 1 (CTR1) and 2 (CTR2). The glutathione S-transferases (GSTs), mu1 (GSTM1), theta1 (GSTT1), and Pi1 (GSTP1) conjugate CDDP with glutathione (GS) and enable its elimination. The DNA lesion induced by CDDP can be removed through the nucleotide excision repair (NER) pathway, mediated by the xeroderma pigmentosum (*XPC, XPD*, and *XPF*) and excision repair cross-complementation group 1 (*ERCC1*) genes, as well as by the mismatch repair (MMR) pathway mediated through proteins encoded by MutL homolog 1, 2, and 3 (*MLH1*, *MSH2*, and *MSH3*, respectively) and exonuclease 1 (*EXO1*) genes. If the repair is not effective, the apoptosis of cells is mediated by proteins encoded by the *TP53*, *CASP8*, *CASP9*, *CASP3*, Fas cell surface death receptor (*FAS*), and Fas ligand (*FASL*) tumor necrosis factor genes. CDDP efflux is mediated via ATPase copper transporters alpha (ATP7A) and beta (ATP7B) (Adapted from Kuo et al. 2007 [22]).

**Figure 2 cancers-15-01759-f002:**
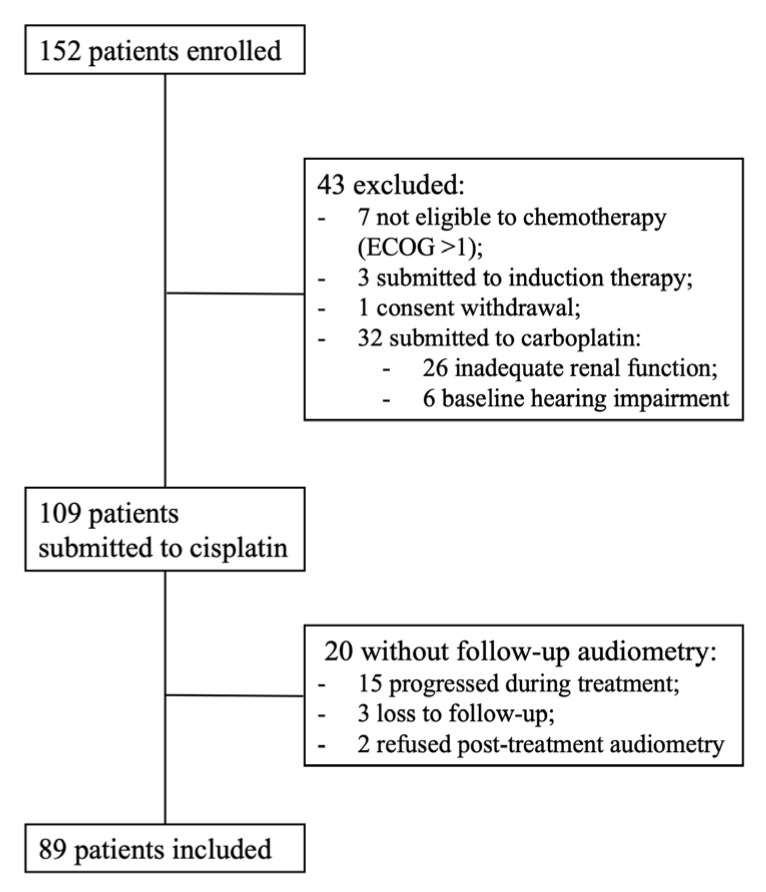
CONSORT diagram for patient selection. *ECOG*: Eastern Cooperative Oncology Group performance status.

**Figure 3 cancers-15-01759-f003:**
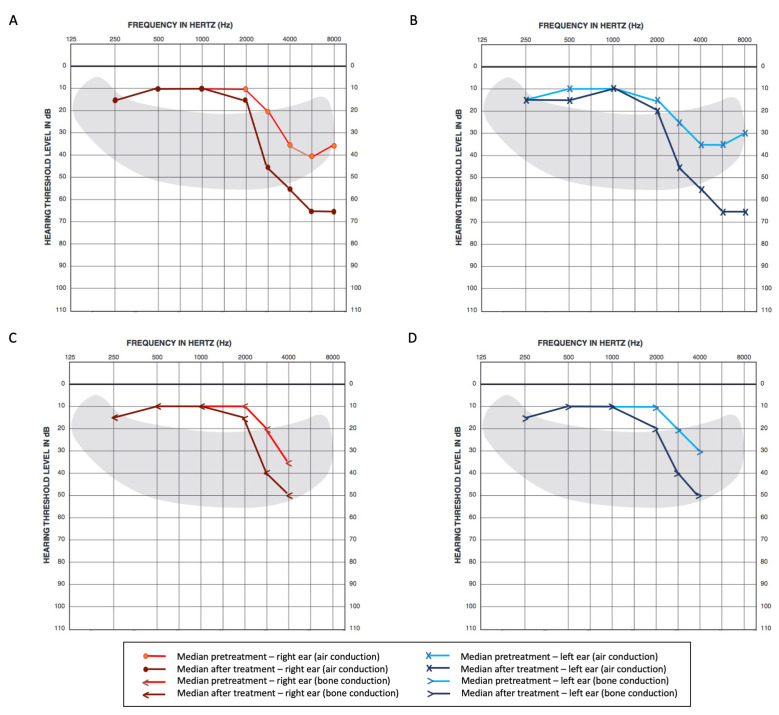
Pure tone thresholds (medians) pre- and post-cisplatin exposure in air and bone conduction audiometry. (**A**) Pure tone thresholds (medians) for 0.25, 1, 2, 3, 4, 6, and 8 kHz in air conduction for the right ear, (**B**) Pure tone thresholds (medians) for 0.25, 1, 2, 3, 4, 6, and 8 kHz in air conduction for the left ear, (**C**) Pure tone thresholds (medians) for 0.25, 1, 2, 3 and 4 kHz in bone conduction for the right ear, (**D**) Pure tone thresholds (medians) for 0.25, 1, 2, 3 and 4 kHz in bone conduction for the left ear.

**Table 1 cancers-15-01759-t001:** Clinical and pathological characteristics of patients included in the study.

Variable	Median (IQR) or N (%)
Age (years)	56 (37–69)
Sex	
Male	82 (92.1)
Female	7 (7.9)
Race (non-white)	9 (10.1)
Tobacco consumption	
Smokers	87 (97.7)
Non-smokers	2 (2.3)
Alcohol consumption	
Active	81 (91)
Abstainers	8 (9)
ECOG performance status	
0	58 (63.5)
1	31 (36.5)
Comorbidities	
BMI	19.4 (13.7–27.5)
Diabetes	9 (10.1)
Hypertension	24 (26.9)
Tumor location	
Oral cavity or oropharynx	55 (61.8)
Hypopharynx or larynx	34 (38.2)
Tumor side	
Right	37 (42.0)
Left	42 (47.7)
Bilateral/medial	9 (10.2)
Histological grade	
Well or moderately differentiated	73 (82.0)
Poorly or undifferentiated	16 (18.0)
Tumor stage	
I or II	6 (6.7)
III or IV	83 (93.3)
Cumulative cisplatin dose (mg/m^2^)	260 (160–300)
Radiotherapy technique (2D)	88 (98.8)
Radiotherapy dose (Gy)	
Supraclavicular fossa	50 (44–50)
Facial (right and left)	44 (44–44)
Boost (right and left)	20 (20–20)

*IQR:* interquartile range; *ECOG:* Eastern Cooperative Oncology Group status performance; *BMI:* body mass index; *Gy:* Gray.

**Table 2 cancers-15-01759-t002:** Hearing thresholds for each frequency in pure tone audiometry.

Frequency (kHz)		Right Ear					Left Ear			
	Pre-Treatment	Post-Treatment	Difference	*p*-Value	BH *p*-Value	Pre-Treatment	Post-Treatment	Difference	*p*-Value	BH *p*-Value
	Median (IQR)	Median (IQR)				Median (IQR)	Median (IQR)			
Air conduction										
0.25	15 (10–20)	15 (10–20)	0 (−5–+5)	0.97	0.97	15 (5–20)	15 (10–20)	0 (−5–+5)	0.10	0.11
0.5	10 (5–15)	10 (5–15)	0 (−5–+5)	0.40	0.45	10 (5–15)	15 (10–20)	0 (−5–+5)	0.06	0.09
1	10 (5–15)	10 (5–15)	0 (−5–+5)	0.05	0.07	10 (5–20)	10 (5–20)	0 (−5–+5)	0.11	0.11
2	10 (5–20)	15 (10–35)	5 (0–15)	<0.001	<0.001	15 (5–20)	20 (10–35)	5 (0–20)	<0.001	<0.001
3	20 (10–35)	45 (20–60)	10 (0–25)	<0.001	<0.001	25 (10–35)	45 (20–60)	10 (0–30)	<0.001	<0.001
4	35 (20–50)	55 (35–65)	15 (0–25)	<0.001	<0.001	35 (20–45)	55 (45–65)	10 (5–30)	<0.001	<0.001
6	40 (20–55)	65 (45–75)	15 (5–35)	<0.001	<0.001	35 (20–55)	65 (50–75)	20 (5–35)	<0.001	<0.001
8	35 (15–55)	65 (50–75)	20 (10–35)	<0.001	<0.001	30 (15–55)	65 (55–75)	30 (10–40)	<0.001	<0.001
Bone conduction										
0.25	15 (10–20)	15 (10–20)	0 (−5–+5)	0.90	0.99	15 (5–15)	15 (10–20)	0 (−5–+5)	0.03	0.05
0.5	10 (5–15)	10 (5–15)	0 (−5–+5)	0.99	0.99	10 (5–15)	10 (5–15)	0 (−5–+5)	0.13	0.15
1	10 (5–15)	10 (5–15)	0 (−5–+5)	0.15	0.23	10 (5–15)	10 (5–20)	0 (−5–+5)	0.16	0.15
2	10 (5–20)	15 (10–35)	5 (0–10)	<0.001	<0.001	10 (5–20)	20 (10–35)	5 (0–20)	<0.001	<0.001
3	20 (10–30)	40 (20–55)	10 (0–25)	<0.001	<0.001	20 (10–35)	40 (20–55)	10 (0–25)	<0.001	<0.001
4	30 (15–45)	50 (30–60)	10 (0–30)	<0.001	<0.001	30 (20–45)	50 (40–60)	10 (0–25)	<0.001	<0.001

*BH:* Benjamini–Hochberg correction by false discovery rate (according to ear side and conduction modality); *IQR:* interquartile range; *kHz:* kilo Hertz.

**Table 3 cancers-15-01759-t003:** Pure tone audiometry, speech audiometry, and hearing classifications before and after cisplatin chemoradiation treatment.

Variable	Pre-Treatment	Post-Treatment	Difference	*p*-Value
	Median (IQR) or N (%)	Median (IQR) or N (%)	Median (IQR) or N (%)	
Pure tone averages				
ISO average (0.5, 1, 2, and 4 kHz)				
*Air conduction (dB)*				
Right ear	17.5 (8–83)	22.5 (8.7–48.7)	5 (−6.25–22.5)	<0.001
Left ear	18.75 (5–43.7)	25 (6.2–75)	6.25 (−5–30)	<0.001
*Bone conduction (dB)*				
Right ear	16.25 (6.2–37.5)	22.5 (10–45)	6.25 (−7.5–20)	<0.001
Left ear	16.25 (5–37.5)	22.5 (6.2–52.5)	6.25 (−5–20)	<0.001
High-frequency averages (3, 4, 6, and 8 kHz)				
*Air conduction (dB)*				
Right ear	35 (0.25–8)	54 (0.25–8)	18 (−51–8)	<0.001
Left ear	34 (0.25–8)	55 (0.25–8)	19 (−54–4)	<0.001
Speech audiometry				
SRT (dB)				
Right ear	10 (5–30)	15 (5–35)	0 (−10–15)	0.12
Left ear	15 (5–35)	15 (5–45)	0 (−10–25)	0.30
SDS (%)				
Right ear	96 (88–100)	92 (80–100)	−4 (−12–4)	0.001
Left ear	96 (72–100)	92 (72–100)	0 (−16–12)	0.06
GBD hearing loss classification				
*Air conduction*				
No loss	57 (64)	32 (35.9)		<0.001
Unilateral	1 (1.1)	4 (4.5)		
Mild	28 (31.4)	37 (41.5)		
Moderate	3 (3.4)	16 (17.9)		
*Bone conduction*				
No loss	62 (69.6)	39 (43.8)		<0.001
Unilateral	1 (1.1)	2 (2.2)		
Mild	23 (25.8)	38 (42.7)		
Moderate	3 (3.4)	10 (11.2)		

*IQR:* interquartile range; *ISO:* International Organization for Standardization; *dB:* decibel; *N:* number; *STR:* speech recognition threshold; *SDS:* speech discrimination score; *GBD:* Global Burden of Disease.

**Table 4 cancers-15-01759-t004:** Significant associations of single nucleotide variants with high-frequency average thresholds (3, 4, 6, and 8 kHz) related to cisplatin-based chemoradiation.

Variable	N		Right Ear			Left Ear	
		Pre-Treatment	Post-Treatment	Difference	Pre-Treatment	Post-Treatment	Difference
		Median (IQR)	Median (IQR)		Median (IQR)	Median (IQR)	
*XPC* c.2815A>C							
AA	34	26.3 (16.9–42.2)	60.0 (43.4–67.8)	23.8 (10.0–39.4)	25.6 (18.8–46.3)	57.5 (40.0–68.8)	27.5 (8.8–40.3)
AC or CC	55	35.0 (20.0–51.3)	58.8 (32.5–68.8)	17.5 (5.0–22.5)	35.0 (20.0–48.8)	55.0 (41.3–65.0)	16.3 (6.3–26.3)
*p*-value			0.008			0.04	
PA			60.5			49.6	
*GSTP1* c.313A>G *+ XPC* c.2815A>C							
AA + AC or CC	25	28.8 (18.8–46.9)	50.0 (30.0–65.6)	17.5 (7.5–23.8)	33.8 (17.5–42.5)	52.5 (38.1–62.5)	16.3 (8.8–23.8)
AG or GG + AA	19	20.0 (15.0–35.0)	61.3 (50.0–67.5)	30.0 (16.3–48.8)	23.8 (13.8–41.3)	62.5 (42.5–68.8)	38.8 (16.3–52.5)
*p*-value			0.005			0.01	
PA			88.0			76.0	
*GSTM1 + EXO1* c.1762G>A							
Present + GG	13	21.3 (18.8–48.1)	31.3 (25.6–63.8)	5.0 (2.5–18.8)	23.8 (17.5–48.1)	38.8 (28.8–60.0)	8.8 (5.0–15.0)
Null + GA or AA	27	37.5 (21.3–50.0)	62.5 (57.5–68.8)	21.3 (16.3–30.0)	40.0 (25.0–48.8)	61.3 (52.5–68.8)	22.5 (11.3–28.8)
*p*-value			0.008			0.005	
PA			75.0			85.0	

*N:* number of patients; *IQR:* interquartile range; *PA:* power analysis. Linear regression with audiometric patterns was adjusted for cumulative cisplatin dose.

**Table 5 cancers-15-01759-t005:** Significant associations for single nucleotide variants and hearing impairment (according to NCI CTCAE v4.03 criteria) related to cisplatin-based chemoradiation.

Variable	N			Ototoxicity		
		G0–G2	G3–G4	OR (95% CI)	*p*-Value	PA (%)
*GSTP1* c.313A>G						
AA	40	33 (52.4)	7 (26.9)	Reference	0.01 ^1^	65
AG or GG	49	30 (47.6)	19 (73.1)	4.20 (1.34–13.16)		
*XPC* c.2815A>C						
AC or CC	55	45 (71.4)	10 (38.5)	Reference	0.01 ^2^	65
AA	34	18 (28.6)	16 (61.5)	3.13 (1.27–7.70)		
*GSTM1 + XPC* c.2815A>C						
Present + AC or CC	22	19 (70.4)	3 (27.3)	Reference	0.02 ^3^	99
Null + AA	16	8 (29.6)	8 (72.7)	8.19 (1.28–52.20)		
*GSTP1* c.313A>G *+ XPC* 2815A>C						
AA + AC or CC	25	24 (72.7)	1 (9.1)	Reference	0.004 ^4^	97
AG or GG + AA	19	9 (27.3)	10 (90.9)	32.22 (3.09–335.52)		
*GSTP1* c.313A>G *+ XPD* c.934G>A						
AA + GG or GA	37	30 (96.8)	7 (63.6)	Reference	0.02 ^5^	92
AG or GG + AA	5	1 (3.2)	4 (36.4)	19.44 (1.59–237.72)		
*GSTP1* c.313A>G *+ EXO1* c.1762G>A						
AA + GG or GA	37	31 (93.9)	6 (54.5)	Reference	0.01 ^6^	81
AG or GG + AA	7	2 (6.1)	5 (45.5)	12.08 (1.60–91.01)		
*XPC* c.2815A>C + *MSH3* c.3133A>G						
AC or CC + AG or GG	23	21 (65.6)	2 (22.2)	Reference	0.009 ^7^	88
AA + AA	18	11 (34.4)	7 (77.8)	17.09 (2.02–144.32)		
*XPC* c.2815A>C + *FASL* c.-844C>T						
AC or CC + TT	12	11 (50.0)	1 (7.7)	Reference	0.01 ^8^	82
AA + CC or CT	23	11 (50.0)	12 (92.3)	22.29 (1.79–276.99)		
*EXO1* c.1762G>A + *P53* c.215G>C						
GG + GG or GC	31	24 (96.0)	7 (58.3)	Reference	0.01 ^9^	85
GA or AA + CC	6	1 (4.0)	5 (41.7)	20.97 (1.66–264.08)		

NCI CTCAE: National Cancer Institute Criteria for Adverse Events; N: number of patients; PA: power analysis; G: grade, OR: odds ratio; CI: confidence interval; IQR: interquartile range. Logistic multivariate regression in hearing impairment (ototoxicity) was adjusted by race and body mass index. ^1^ p bootstrap = 0.01; ^2^ p bootstrap = 0.007; ^3^ p bootstrap = 0.01; ^4^ p bootstrap = 0.009; ^5^ p bootstrap = 0.002; ^6^ p bootstrap = 0.005; ^7^ p bootstrap = 0.001; ^8^ p bootstrap = 0.005; ^9^ p bootstrap = 0.005.

## Data Availability

The data are not publicly accessible due to privacy and ethical restrictions but can be made available upon request from the corresponding author.

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
