# Peer review of "Association of Clinical Aspects and Genetic Variants with the Severity of Cisplatin-Induced Ototoxicity in Head and Neck Squamous Cell Carcinoma: A Prospective Cohort Study"

_cancers, 2023, doi:10.3390/cancers15061759_

Round 1

Reviewer 1 Report

The manuscript by Macedo et al. provides some evidence on genetic variants and increased risk of cisplatin-induced ototoxicity. Other previous studies tried to identify the link between other genetic variants like MATE1, ABCC2, COMT and platinum ototoxicity. The authors looked at SNVs of mainly DNA damage repair and apoptosis key genes, which are certainly involved in response to cisplatin-induced DNA adducts.  

Minor comments:

In Figure 3, the median pre treatment and post treatment should be plotted using more different contrast  colours to be easily distinguished by the readers.

The p-value in table 2 should be corrected to adjusted p-value using Bemjamini and Hochberg method and plotted in the table. 

Author Response

Dear Reviewer,

Please find the revised version of the manuscript for your considerations. We believe your suggestions were clear and useful.

We have addressed the following comments:

Reviewer 1 suggestion 1:

In Figure 3, the median pre treatment and post treatment should be plotted using more different contrast colours to be easily distinguished by the readers.

Authors’ response:

We appreciate your suggestion, and we have changed the colours in Figure 3 (Results, Page 8).

Reviewer 1 suggestion 2:

The p-value in table 2 should be corrected to adjusted p-value using Bemjamini and Hochberg method and plotted in the table.”

Authors’ response:

As requested, we have included the adjustments according to Benjamini and Hochberg (Benjamini, Y., & Hochberg, Y. (1995). Controlling the False Discovery Rate: A Practical and Powerful Approach to Multiple Testing. Journal of the Royal Statistical Society: Series B, Methodological, 57(1), 289–300. doi:10.1111/j.2517-6161.1995.tb02031.x) in Table 2 (Results, Page 9). We added additional details on Methods, Page 5, line 289. We hope these changes enhanced the quality of the manuscript.

We genuinely appreciate your suggestions. Thank you in advance.

Reviewer 2 Report

In this manuscript, Macedo et al correlated the Clinicopathological features with genetic variants encoding glutathione S-transferases (GSTT1, GSTM1, GSTP1), nucleotide excision repair (XPC, XPD, XPF, ERCC1), mismatch repair (MLH1, MSH2, MSH3, EXO1), and apoptosis (P53, CASP8, CASP9, CASP3, FAS, FASL) related proteins were analyzed regarding ototoxicity. They correlated the treatment-related hearing loss with the genotypes of the SNPs. However, there are several major comments, which need to be resolved before considering it for publication.

a) It is not clear how authors selected the specific SNPs of the selected genes. Because one gene may have several 100 to thousands of SNP. Moreover, one SNP may be correlated with several other SNP with LD. Please clarified it in the introduction, method, and discussion sections.

b) As authors are more interested in hearing loss after treatment, the author should study the genetic variants of genes associated with the hearing process.

c) Please show some functional aspects of the SNPs associated with hearing loss.

d) Are those SNPs located in the coding or noncoding region? If SNPS are at the coding region, give the coding change of the protein. If the regulating SNPs are located in the noncoding regions, generally they are located in enhancer or suppressor regions. Please check the activity of the SNP by luciferase assay.

e) Similar genotype-specific experiments have already been published earlier. It is better to show some mechanistic and functional aspects of those SNPs.

f) As per recent studies, Stress is the most important factor, which could induce tumorigenesis and promote cancer development via modulation of genetic and epigenetic changes. Moreover, it could modulate the therapeutic efficacy via the regulation of differential gene expressions (https://doi.org/10.3389/fonc.2020.01492;  https://doi.org/10.1038/s41568-021-00395-5). So stress or other physiological condition will largely influence the effect of CDDP treatment. Though the authors are showing hypertension in table -1, the author should stratify their genotype data with the earlier or present stress levels of the patients.

On the other hand, lots of recent research already showed that stress management via meditation related to mind body and breath could help to get better treatment outcomes. (https://doi.org/10.1073/pnas.2110455118; https://doi.org/10.1186/s13063-018-3103-8; doi: 10.3389/fpsyg.2021.635816; DOI 10.1007/s11764-012-0252-8, https://doi.org/10.1016/j.urolonc.2020.09.011; doi:10.1002/nur.22169 ). Therefore, the author could include the noninvasive procedure and analyze the outcome with the genetic variations.

 g) Please show the survival analysis curve.

h) Authors are overestimating their results. Please revise conclusion

Author Response

Dear reviewer,

Thank you for your considerations on the manuscript. Please find below our comments:

“In this manuscript, Macedo et al correlated the Clinicopathological features with genetic variants encoding glutathione S-transferases (GSTT1, GSTM1, GSTP1), nucleotide excision repair (XPC, XPD, XPF, ERCC1), mismatch repair (MLH1, MSH2, MSH3, EXO1), and apoptosis (P53, CASP8, CASP9, CASP3, FAS, FASL) related proteins were analyzed regarding ototoxicity. They correlated the treatment-related hearing loss with the genotypes of the SNPs. However, there are several major comments, which need to be resolved before considering it for publication.”

Reviewer 2 suggestion 1:

“a) It is not clear how authors selected the specific SNPs of the selected genes. Because one gene may have several 100 to thousands of SNP. Moreover, one SNP may be correlated with several other SNP with LD. Please clarified it in the introduction, method, and discussion sections.”

Authors’ response:

In the revised version of the manuscript, we included a flowchart representing the selection process (Figure A1 - Page 18), and described the selection criteria in Methods, Page 5, Line 265. As not all SNVs on genes related to CDDP metabolism, DNA repair and apoptosis of damaged cells were selected for this study, we included this information as a limitation in the Discussion section (Page 15, Line 708). We hope that this information can make the manuscript clearer for readers.

Reviewer 2 suggestion 2:

“b) As authors are more interested in hearing loss after treatment, the author should study the genetic variants of genes associated with the hearing process.”

Authors’ response:

            Thank you for your impression. Certainly, ototoxicity related to cisplatin may not the only possible explanation for hearing loss during treatment. For this reason, we highlight the lack of other studied SNVs involved in ototoxicity unrelated to cisplatin metabolism as potential confounding factors not addressed in our analysis (Introduction section, Page 3, Lines 129-132; Discussion section, Page 15, Line 715). In this revised version, we included more data on genes related to hearing loss, identifying this as a limitation to our study.

Reviewer 2 suggestions 3 and 4:

“c) Please show some functional aspects of the SNPs associated with hearing loss.”

“d) Are those SNPs located in the coding or noncoding region? If SNPS are at the coding region, give the coding change of the protein. If the regulating SNPs are located in the noncoding regions, generally they are located in enhancer or suppressor regions. Please check the activity of the SNP by luciferase assay.”

Authors’ response:

We were also concerned with the description of functional characteristics of each single nucleotide variant (SNV). For this reason, we had previously included a concise description of the SNVs activities available (Table A1, Page 17). In this revised version, we have included the suggested details on table A1, adding further information related to assays previously performed and reported in current literature, as well as SNV locations. We hope these changes enhance the quality of the study. We appreciate your suggestion.

Reviewer 2 suggestion 5:

“e) Similar genotype-specific experiments have already been published earlier. It is better to show some mechanistic and functional aspects of those SNPs.”

Authors’ response:

            Apart from the revision of Table A1, Page 17, we also included further details on the functional aspects of the SNVs related to hearing loss in the Discussion section (Page 14, Line 625; Page 15, Line 675; Page 15, Line 688)

Reviewer 2 suggestion 6:

“f) As per recent studies, Stress is the most important factor, which could induce tumorigenesis and promote cancer development via modulation of genetic and epigenetic changes. Moreover, it could modulate the therapeutic efficacy via the regulation of differential gene expressions (https://doi.org/10.3389/fonc.2020.01492;  https://doi.org/10.1038/s41568-021-00395-5). So stress or other physiological condition will largely influence the effect of CDDP treatment. Though the authors are showing hypertension in table -1, the author should stratify their genotype data with the earlier or present stress levels of the patients. On the other hand, lots of recent research already showed that stress management via meditation related to mind body and breath could help to get better treatment outcomes. (https://doi.org/10.1073/pnas.2110455118; https://doi.org/10.1186/s13063-018-3103-8; doi: 10.3389/fpsyg.2021.635816; DOI 10.1007/s11764-012-0252-8, https://doi.org/10.1016/j.urolonc.2020.09.011; doi:10.1002/nur.22169). Therefore, the author could include the noninvasive procedure and analyze the outcome with the genetic variations.”

Authors’ response:

Unfortunately, by the time this cohort was planned, the evidence related to stress as a potential risk was not available. Therefore, distress scores were not performed and included in this study longitudinally. A significant proportion of patients are currently deceased, limiting the possibility of any assessment. However, we have included some considerations in the Discussion section, Page 14, Line 615 and Page 15, Line 723, prompting for further investigation on this matter. Thank you for the suggestion.

Reviewer 2 suggestion 7:

“g) Please show the survival analysis curve.”

Authors’ response:

Indeed, efficacy may also be influenced by the mentioned SNVs in the study, and our group has prepared data on survival. In this specific manuscript, however, we focused on ototoxicity and gathering a detailed description on hearing loss related to cisplatin. Therefore, we did not present data regarding treatment efficacy as tumor response and survival, since data related to efficacy, including survival analysis, is currently under review for publication in a separate article.

Reviewer 2 suggestion 8:

“h) Authors are overestimating their results. Please revise conclusion”

Authors’ response:

We have reformulated the conclusion, in Page 16, Line 776. We hope the revised version is more suitable.

Reviewer 3 Report

This is nice research work about the ototoxic effect of CDDP treatment in HNSCC. It is presented clearly, concise and with scientific soundness. 

Author Response

We would like to thank the reviewer for his/her kind remarks.

Round 2

Reviewer 2 Report

The manuscript has been revised satisfactorily. The pattern of study is very old. It is better to update the lab procedures including recent updates in the field.